# General simulation method for spontaneous parametric down- and parametric up-conversion experiments

Felix Riexinger[1,2]*, Mirco Kutas[1,2], Björn Haase[1,2], Patricia Bickert[1], Daniel Molter[1], Michael Bortz[1], and Georg von Freymann[1,2]

**1** Fraunhofer Institute for Industrial Mathematics ITWM, Fraunhofer-Platz 1, 67663 Kaiserslautern, Germany
**2** Department of Physics and Research Center OPTIMAS, Technische Universität Kaiserslautern (TUK), 67663 Kaiserslautern, Germany
* felix.riexinger@itwm.fraunhofer.de

March 8, 2022

## 1 Abstract

**Spontaneous parametric down-conversion (SPDC) sources are an important technology for quantum sensing and imaging. We demonstrate a general simulation method, based on modeling from first principles, reproducing the spectrally and spatially resolved absolute counts of a SPDC experiment. By additionally simulating parametric up- and down-conversion processes with thermal photons as well as effects of the optical system we accomplish good agreement with the experimental results. This method is broadly applicable and allows for the separation of contributing processes, virtual characterization of SPDC sources, and enables the simulation of many quantum based applications.**

## 1 Introduction

Entangled and correlated photon pairs have become the basis for many applications in quantum optics. They are used in various quantum based schemes such as ghost imaging [1, 2], optical coherence tomography [3], spectroscopy [4, 5], quantum sensing [6], and imaging with undetected photons [7–9]. One of the most prominent methods for generating entangled photon pairs is spontaneous parametric down-conversion (SPDC), where a pump laser photon decays into two lower-energy photons in a medium with a second-order nonlinearity. The entanglement can exist in many of the photon properties, such as polarization, wavelength, or momentum, making SPDC a versatile source for entangled photons. Additionally, the process is easy to implement and well understood experimentally. However, an accurate simulation method for SPDC sources is required as the foundation for simulations of many applications with entangled photons. The availability of detailed simulations can help resolve problems such as finding limitations to the resolution or visibility of quantum imaging [8, 9]. This issue is becoming more relevant as applications for SPDC sources are on the verge of a breakthrough, but their limiting factors need to be understood better to exploit the full potential of such quantum based applications.

The theory of SPDC is well developed and multiple approaches to simulating the properties of the created photons are available [10, 11]. Existing SPDC simulations are tailored for specialized applications limiting their general applicability. The limitations include restriction to the paraxial regime [11, 12] and narrow frequency or wave vector spreads [12–14]. Many works do not predict absolute photon conversion rates [12–16]. In this letter we propose and demonstrate a novel simulation method for SPDC sources and the subsequent measurement

setup. The sparse use of approximations makes the underlying model applicable to a wide range of SPDC sources from the ultraviolet to the terahertz regime.

Our method reproduces the spectrally and spatially resolved absolute photon count rates. This is demonstrated on an experiment with idler photons in the terahertz range and signal photons in the visible range. The extreme wavelength spread between signal and idler leads to a setup that covers a large range in frequency and emission directions and further has multiple quasi-phasematching (QPM) orders overlapping in the same wavelength range. In the terahertz range additional processes such as parametric up- and (nonspontaneous) down-conversion occur parallel to SPDC. In order to adequately match the experimental data we include these processes in our simulation. The high qualitative and quantitative agreement with experimental results demonstrates the capabilities of our simulation method even for complex SPDC sources.

## 2   Theory

Our model is based on the second-order nonlinear interaction of electromagnetic fields together with a first-order perturbation theory approximation. We start with the Hamiltonian [17]

$$H_{\text{NL}}(t) = \frac{1}{3}\varepsilon_0 \int d\mathbf{r}\, \zeta^{(2)}_{jkl}(\mathbf{r})\mathbf{D}_j(\mathbf{r},t)\mathbf{D}_k(\mathbf{r},t)\mathbf{D}_l(\mathbf{r},t), \tag{1}$$

where $\zeta^{(2)}_{jkl}$ is the second-order inverse susceptibility tensor and $\mathbf{D}$ are the displacement fields. This formulation is necessary to ensure consistency after quantization [18, 19].

We describe the pump beam as a classical monochromatic Gaussian beam with linear polarization. We assume that the pump is undepleted. In addition, we use the approximation of a collimated beam such that the curvature and the Gouy phase can be neglected. The pump propagates along the z-axis, which we define parallel to the optical axis of the system. The signal and idler fields are described using a plane-wave decomposition separated into positive and negative frequency components $\hat{\mathbf{D}}^+$ and $\hat{\mathbf{D}}^-$ with

$$\hat{\mathbf{D}}^+(\mathbf{r},t) = \sum_{\mathbf{k},\sigma} i\sqrt{\frac{\varepsilon_0 n_{\mathbf{k}}^2 \hbar \omega_{\mathbf{k}}}{2V}}\, \hat{a}_{\mathbf{k},\sigma}\, \epsilon_{\mathbf{k},\sigma}\, e^{i(\mathbf{k}\cdot\mathbf{r}+\omega_{\mathbf{k}} t)}, \tag{2}$$

and $\hat{\mathbf{D}}^-$ being the hermitian conjugate of $\hat{\mathbf{D}}^+$. Here, $V$ is the quantization volume, $\hat{a}_{\mathbf{k},\sigma}$ is the annihilation operator for a photon with momentum $\mathbf{k}$, and $\epsilon_{\mathbf{k},\sigma}$ is the direction of the displacement field vector indexed with the polarization $\sigma$.

We then approximate the two-photon state $|\psi(t)\rangle$ using first-order perturbation theory. Under the assumption that the quantization volume is large, we can make a transition from sums to integrals in Eq. (2). From this we obtain the signal count rate density:

$$\Gamma_{\text{d}}(\mathbf{k}_{\text{s}}) = \frac{1}{T_{\text{I}}}\langle\psi(T_{\text{I}})|\hat{a}^\dagger(\mathbf{k}_{\text{s}})\hat{a}(\mathbf{k}_{\text{s}})|\psi(T_{\text{I}})\rangle \tag{3}$$

$$= Z \sum_{\sigma_{\text{s}},\sigma_{\text{i}}}\sum_{m \text{ odd}} \int dk_{\text{i}}^3 \|A(\mathbf{k}_{\text{s}},\mathbf{k}_{\text{i}})\|^2, \tag{4}$$

with

$$A(\mathbf{k}_{\text{s}},\mathbf{k}_{\text{i}}) = \frac{\chi^{(2)}_{\text{eff}}}{m}\sqrt{\frac{\omega_{\text{s}}\omega_{\text{i}}}{n_{\text{s}}^2 n_{\text{i}}^2}}\,\text{sinc}\left[\frac{1}{2}\Delta k_z L\right]\exp\left[-\frac{1}{4}(\Delta k_x^2 + \Delta k_y^2)w_{\text{p}}^2\right]\text{sinc}\left[\frac{1}{2}\Delta\omega T_{\text{I}}\right]$$

and

$$Z = \frac{16 P w_{\mathrm{p}}^2 L^2 T_I}{(2\pi)^7 \varepsilon_0 n_{\mathrm{p}} c}. \tag{5}$$

Here $T_{\mathrm{I}}$ is the interaction time for a pump photon with the nonlinear medium, $m$ denotes the odd QPM orders, and we sum over permutations of the indices $j$, $k$, $l$ and substitute new indices to denote pump (p), signal (s) and idler (i). $\Delta k_z = \mathbf{k}_{\mathrm{p}z} - \mathbf{k}_{\mathrm{s}z} - \mathbf{k}_{\mathrm{i}z} + k_\Lambda$ and $\Delta k_j = \mathbf{k}_{\mathrm{s}j} + \mathbf{k}_{\mathrm{i}j}$ with $j = x, y$ are the longitudinal and transversal phase mismatches, and $\Delta\omega = \omega_{\mathrm{p}} - \omega_{\mathrm{s}} - \omega_{\mathrm{i}}$ corresponds to the energy mismatch. The width of the transverse part is determined by the waist radius $w_{\mathrm{p}}$ of the pump beam. The periodic poling offset $k_\Lambda = 2m\pi/\Lambda$ depends on the poling period of the crystal $\Lambda$ and the QPM order $m$. $P$ denotes the power of the pump beam and $L$ the length of the crystal. The refractive indices $n$ and $\chi_{\mathrm{eff}}^{(2)}$ are functions of the wave vectors $\mathbf{k}_{\mathrm{p}}$, $\mathbf{k}_{\mathrm{s}}$, $\mathbf{k}_{\mathrm{i}}$ and their corresponding polarizations. The spectral dependence of the nonlinear coefficient is modeled with Miller's rule [20]. The spatial variation of $\chi_{\mathrm{eff}}^{(2)}$ is considered by calculating the effective value from the tensor components and displacement field directions [21]. Spatial variation in the refractive indices is considered following [22]. The spectral dependencies are especially relevant since we simulate a large spectral range in the terahertz regime, from $\sim 0.1$ THz to $3.6$ THz. The spatial variation cannot be neglected here, because the transverse momentum conservation dictates a large emission angle range for the idler photons.

With the idler in the terahertz range, at room temperature, thermal photons at the idler wavelength have to be taken into account [23]. These thermal photons interact with the pump laser as well. Through parametric down-conversion, additional photons at the signal wavelength are created. We derive this process analogously to SPDC. Instead of an initial vacuum state we have a thermal state for the idler. This leads to the same phasematching properties, but instead of a "1" from the vacuum fluctuations we obtain the thermal fluctuations

$$N_{\mathrm{th}} = \frac{1}{\exp(\hbar\omega_{\mathrm{i}}/k_{\mathrm{B}}T_{\mathrm{c}}) - 1}, \tag{6}$$

where $T_{\mathrm{c}}$ is the temperature of the crystal. The influence of thermal fluctuations for the signal can be neglected.

The emitted spectrum is propagated from the crystal onto a detector through an optical system. The number of counts for a single detector pixel is obtained by integrating over all signal wave vectors that are propagated to this pixel. However, not all photons created in the crystal arrive at the detector. We consider the losses from internal reflection in the crystal, during propagation through the optical setup and the efficiency of the detector summarized into a single factor $\eta$. With this we obtain the photon counts for the detector pixel with indices $(i, j)$

$$R_{\mathrm{d}}^{(i,j)} = \eta T (1 + N_{\mathrm{th}}) \int_{\Omega(i,j)} \mathrm{d}\mathbf{k}_{\mathrm{s}} \Gamma_{\mathrm{d}}(\mathbf{k}_{\mathrm{s}}), \tag{7}$$

where $T$ is the illumination time of the detector and $\Omega(i, j)$ is defined such that every ray with $\mathbf{k}_\Omega \in \Omega(i, j)$ at the crystal exit is propagated through the optical system onto the detector pixel $(i, j)$. Along the same lines we obtain the rate for the parametric up-conversion process, where a thermal photon and a laser photon merge into a signal photon. The only changes are the signs of the terahertz frequency $\omega_{\mathrm{i}}$ in the terms $\Delta\omega$ and the wave vector $\mathbf{k}_{\mathrm{i}j}$ in the respective $\Delta k_j$ terms with $j = x, y, z$. Up-conversion is only caused by thermal fluctuations, such that the up-conversion rate is given by:

$$R_{\mathrm{u}}^{(i,j)} = \eta T N_{\mathrm{th}} \int_{\Omega(i,j)} \mathrm{d}\mathbf{k}_{\mathrm{s}} \Gamma_{\mathrm{u}}(\mathbf{k}_{\mathrm{s}}). \tag{8}$$

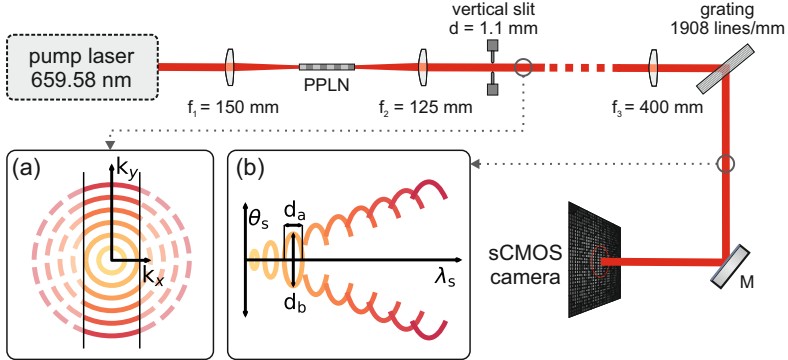

Figure 1: Layout of the experimental setup. Only the components used in the simulations are included. f: lens. PPLN: periodically poled MgO-doped LiNbO$_3$-crystal. M: mirror. Length of optical paths are not to scale. (a) represents a cross section of the spectrum after the slit. (b) sketches the separation of spectral components by the grating. The ratio $d_a/d_b$ quantifies the asymmetry introduced in this step.

## 2.1 Simulated Setup

A sketch of the model for the experimental setup is shown in Fig. 1, where only the simulated components are shown. The setup consists of a narrow bandwidth laser, a nonlinear crystal and an imaging system. The spatial and spectral components of the created signal photons are separated by the imaging system and imaged onto the detector. The imaging system contains a slit to limit the transmission of rays with large $\mathbf{k}_x$, shown in Fig. 1 (a), resulting in a sharper image on the detector. This effect is shown in Fig. 1 (b), where the remaining parts of the ellipses are separated along the $\theta_s$ axis. A transmission grating separates the spectral components of the photons. Photons with a fixed wavelength are emitted in a cone shape, shown in Fig. 1 (a). This leads to an imperfect separation of spatial and spectral components, which is shown by the full and partial ellipses in Fig. 1 (b). The ratio $d_a/d_b$ is a measure of how much an ellipse is squeezed. As it approaches zero, the ellipses are imaged as vertical lines, making the spread in $\theta_s$ large and the one in $\lambda_s$ small. The system further contains several filters and Bragg gratings to suppress the pump radiation that are not considered in this model. A detailed description of the experimental realization is given by Haase *et al.* [24]. The laser is modeled with a wavelength of $\lambda_p = 659.58$ nm and a beamwaist $w_p = 43\ \mu$m. The nonlinear crystal is a periodically poled MgO-doped lithium niobate crystal with dimensions $5 \times 1 \times 10$ mm$^3$ ($H \times W \times L$) and a poling period of 170 $\mu$m. Due to symmetries in the nonlinear susceptibility only the values of the $\chi^{(2)}_{333}$ and $\chi^{(2)}_{311}$ components need to be considered. The $\chi^{(2)}_{222}$ component is not relevant to any processes in our setup. We use a value of $\chi^{(2)}_{333} = 327$ pm/V at 661 nm and 0.75 THz which we obtained from a fit to the experimental results. The fitted $\chi^{(2)}_{333}$ value agrees with a scaled value from the infrared range [25] but is $\sim 1.5$ times larger than other scaled values from the terahertz range [26, 27]. The difference can be explained by different factors for the Hamiltonian ($\frac{1}{2}$ and $\frac{1}{3}$) used in the classical and quantum approaches. Compared to our previous estimation [24] this value is two times larger. There are two reasons for this: First, we did not use Miller's rule in the previous method, but assumed a constant value. Second, we added the full optical model to the simulation, which changes the shape of the spectrum. The value of $\chi^{(2)}_{311} = 49$ pm/V at 661 nm and 0.75 THz is a scaled value from [25]. The contributions of this parameter are small such that a reliable fit is not possible. The refractive indices, for $\sim 5$ mol.% MgO-doped LiNbO$_3$, are taken from [28, 29].

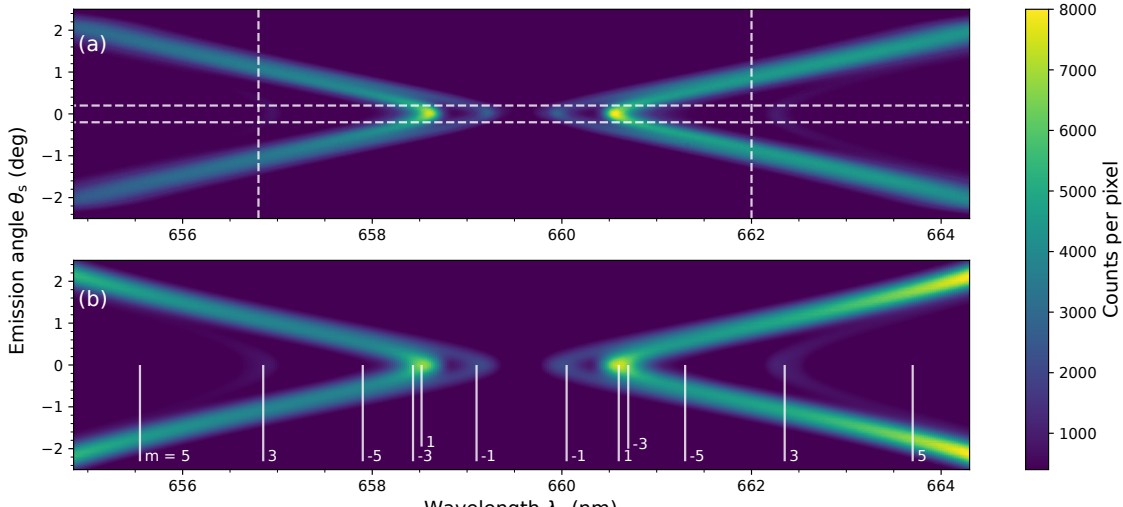

Figure 2: Experimental (a) and simulated (b) frequency-angular spectrum. The average dark count rate is subtracted from the experimental spectrum. The dashed lines in (a) denote the location of the cuts presented in Fig. 3 and 4. The numbers $m$ in (b) indicate the contributions of the different QPM orders. The tails on the left correspond to up-conversion, the ones on the right to down-conversion.

## 2.2 Optical Propagation

The propagation through the optical system is modeled by paraxial ray optics assuming ideal optical components. The paraxial approximation is reasonable as only the signal photons are detected. The idler photons which are emitted at larger angles are not considered. Since the longitudinal positions of the optical components are not exactly known, they are estimated from the imaging properties of the setup. This is done using numerical optimization to minimize the squared difference of three experimentally measured values: the transformation of wavelength into a position on the x-axis, the relation between emission angle and position on the y-axis, and the squeezing of a monochromatic circular beam, defined by the ratio $d_a/d_b$. We also penalize deviations from the measured positions. This procedure of estimating the parameters of the setup mimics the alignment process performed in the experiment, where the components are moved around their nominal positions to obtain a sharper image. The calculated values deviate from the nominal distances of the setup, but reproduce the imaging properties of the actual experiment. Further sources of errors in the optical system such as misalignment transverse to the optical axis or deviations from nominal values are not included in our model.

## 2.3 Numerical Methods

We employ a Monte-Carlo integration scheme to evaluate the integrals in Eq. (3), approximating each squared sinc functions with the sum of three scaled Gaussians. This allows for efficient sampling while maintaining a small error in the approximated function. Other approximations [10, 30] are either less accurate or less efficient. The sum in Eq. (3) is evaluated up to the seventh QPM order. Since positive and negative orders are possible, a total of eight summands are evaluated. Higher orders contribute less than 70 counts per pixel to the spectrum in the observed range and are therefore neglected. We consider both ordinary and extraordinary polarization for signal and idler. Type 0 parametric conversion contributes most of the spectrum in the investigated range, type I contributes less than 165 counts per pixel in

the region above 663.7 nm and less than 2 counts per pixel elsewhere. Type II contributes less than 1 count per pixel in the whole range.

## 3   Results

The experimental and numerical results for the full spectrum are shown in Fig. 2. The experimental spectrum is a single image as recorded by the sCMOS camera with the average dark count rate subtracted. Simulated and experimental spectrum show four distinct tails for up- as well as down-conversion. The contributing QPM orders for each tail are given in panel (b) of the figure. The tails corresponding to the fifth QPM order are barely visible, while the contributions of the seventh order cannot be distinguished at all due to the lower conversion efficiency of higher orders. The experiment is limited to a scattering angle of around $\pm 2.3°$ by the apertures of the optical components. The simulated spectrum extends beyond this angle as the limiting apertures are not included in the model.

The simulation shows more counts than the experiment at wavelengths lower than 655 nm and higher than 663.5 nm for up- and down-conversion, respectively. Potential reasons are limiting apertures in the experiment or an incorrect model for the terahertz refractive indices. The idler photons in this range have a frequency of over 2.5 THz which is beyond the measured range of our reference, covering frequencies from 0.3 THz to 1.9 THz. The value of $\chi_{\text{eff}}^{(2)}$ depends on the refractive indices in our model such that an overestimation of $n$ increases the counts.

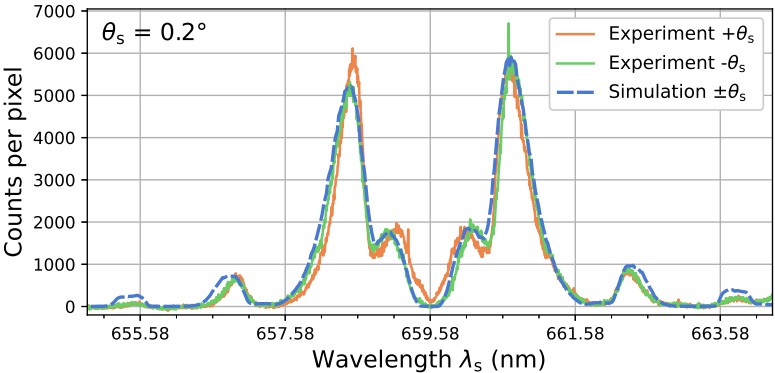

Figure 3: Horizontal cut through the experimental and simulated spectra at $\theta_s = 0.2°$. The positive and negative sign for $\theta_s$ correspond to cuts through the upper and lower half of Fig. 2 respectively. The simulated spectrum is symmetric in $\theta_s$, thus only one line is shown.

Figures 3 and 4 show horizontal and vertical cuts of the spectrum. The positive and negative values for $\theta_s$ correspond to the sign given in Fig. 2. As the simulated spectrum is symmetric in $\theta_s$ only one line is shown. Both cuts show good agreement in position, width and height of the peaks. The experimental spectrum shows some remaining pump light around the laser wavelength at the center of Fig. 3. This is more pronounced for the $+\theta_s$ case. In the model we assume the pump to be blocked completely, therefore the simulated spectrum does not show the pump rest.

The tails of the experimental spectrum are slightly narrower and more pronounced, which is due to the experiment being adjusted for a maximally sharp image, while the simulated setup is not. It is optimized to reproduce three imaging characteristics of the experiment. Observed differences between simulation and experiment are of the same order of magnitude as the

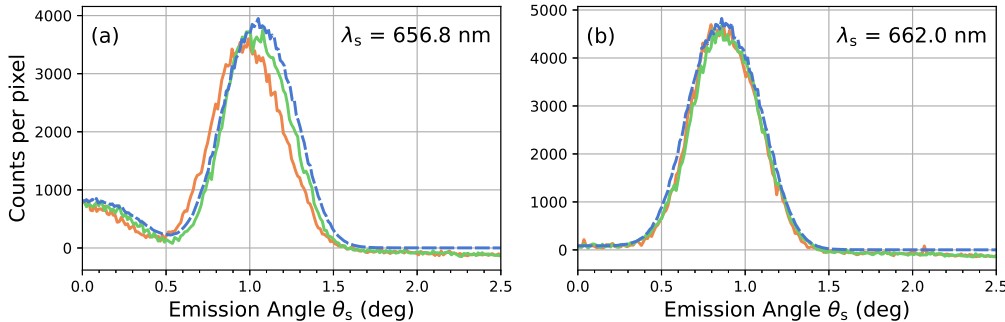

Figure 4: Vertical cuts through the experimental and simulated spectra. Panel (a) shows a cut in the up-conversion regime, panel (b) in the down-conversion regime. The color coding is the same as in Fig. 3.

differences between the two experimental results. Peaks found in the cuts from larger angles or larger wavelengths match in shape and position. The peaks of the simulated spectrum are significantly higher for this region. Note that the simulated spectrum exhibits some minor modulation along the $\theta_s$-axis which are numerical artifacts caused by the employed sampling method.

## 4 Conclusion

The demonstrated simulation of parametric conversion spectra shows good agreement with the experiment in spectral and angular distributions as well as absolute photon counts. Qualitative and quantitative features of the experimentally obtained spectrum can be reproduced. Various effects such as SPDC, the influence of thermal photons and parametric up-conversion were simulated. This shows the potential of applying our model to predict accurate characteristics of photon sources. And the possibility of identifying the contributions of different processes. The model can be simplified for faster computation times. An important step to improve the simulation results is to use better estimates for the crystal characteristics in the investigated frequency range. The method provides significant benefits over traditional methods such as the calculation of phasematching curves as no information about the width or intensity of the spectrum is provided there. Our numerical method also allows for reconstruction of the spectrum at the crystal face which allows to investigate spatial, spectral and correlation properties without the need of building optical setups for measuring them. Compared to previous results [24], the propagation of the spectrum through the measurement setup improves the simulation results. Further research is needed to separate influences of model errors in the up- and down-conversion model from those in the optical measurement setup. Nonetheless, the presented model provides the necessary basis for the simulation of many quantum optic applications such as quantum imaging.

**Funding information** Fraunhofer-Gesellschaft (Lighthouse project QUILT)

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
