# Peer review of "General simulation method for spontaneous parametric down- and parametric up-conversion experiments"

_SciPost Physics_

## Round 1 · Referee Report · Anonymous (Referee 1) · 2022-9-22

Strengths

1) Excellent agreement between the theoretical/numerical results an the experimental data. 2) The paper is very well written and clear in the exposition of the theory and the comparison with experiments. 3) The results are novel enough to deserve publication.

Weaknesses

1) Few of the starting elements overlap with Ref. [24] by some of the same authors 2) A more detailed comparison with Ref. [24] is needed to enucleate better the novelty elements

Report

The research and the presentation are in my view fine and they satisfy the criteria of publication in SciPost. My only comment is that there is a substantial overlap in the presentation of the theory with Ref. [24] of some of the same author. The introduction of the theory/numerics is worth here but the authors have confined a comparison with the results of Ref. [24] to a single line in the present version of the paper: 'Compared to previous results [24], the propagation of the spectrum through the measurement setup improves the simulation results'. This is not sufficient for a reader of SciPost and I invite the authors to significantly expand these considerations either in the conclusions, or, better, at the end of Section 3 by presenting a one-to-one comparison of the results of Section 3 with the results of Ref. [24]. This would help the reader to understand better the elements of novelty, significance and originality of the present paper.

Requested changes

the authors should significantly expand the considerations and comparisons with Ref.[24] either in the conclusions, or, better, at the end of Section 3. they should present a one-to-one comparison of the results of Section 3 with the results of Ref. [24]. This would help the reader to understand better the elements of novelty, significance and originality of the present paper.

---

## Editorial Decision

resubmitted